# Sparks in the dark

Olga Sunneborn Gudnadottir[1], Axel Gallén[1*], Giulia Ripellino[1],
Jochen J. Heinrich[2], Raazesh Sainudiin[3] and Rebeca Gonzalez Suarez[1]

**1** Department of Physics and Astronomy, Uppsala University,
Läderhyggsvägen 1, Uppsala, 752 37, Sweden
**2** Department of Physics, University of Oregon, 120 Willamette Hall,
1371 E 13th Avenue, Eugene, Oregon, 97403, United States.
**3** Department of Mathematics, Uppsala University,
Läderhyggsvägen 1, Uppsala, 752 37, Sweden.

⋆ axel.lars.gallen@cern.ch

## Abstract

This study presents a novel method for the definition of signal regions in searches for new physics at collider experiments. By leveraging multi-dimensional histograms with precise arithmetic and utilizing the SparkDensityTree library, it is possible to identify high-density regions within the available phase space, potentially improving sensitivity to very small signals. Inspired by a search for dark mesons at the ATLAS experiment, CMS open data is used for this proof-of-concept intentionally targeting an already excluded signal. Signal regions are defined based on density estimates of signal and background. These preliminary regions align well with the physical properties of the signal while effectively rejecting background events.

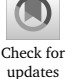

# 1 Introduction

Collider experiments in high-energy physics often deal with large amounts of experimental data. The two general-purpose experiments at CERN's Large Hadron Collider (LHC), ATLAS and CMS, record about 10 PB of data per year. These data are then analyzed for, e.g., consistency with different theoretical models, which involves both isolating a small signal from large background and data-driven corrections to background estimates. A pre-selection of data is performed based on the particles involved in the experimental signature of the signal. Subsequently, the resulting dataset is explored with the objective to create a phase-space region enriched in signal events. This enriched region allows for a statistical analysis that is sensitive to the signal. Optimizing the region involves using theoretical knowledge of the signatures and kinematic behavior of the signal and background processes to define new variables, and a tedious process of exploring the data using 1D or 2D histograms. Machine learning classifiers are also commonly used at this stage, which both hone in on the region without the same need for manual optimization and utilize complex relationships between variables. The downside of these methods is that the interdependence of the variables is never made explicit, and the analysis becomes harder to understand than one defined in terms of intervals in each variable. This matters not only for the understanding of the individual physicist, but also matters for reinterpretations of the results. This paper is a proof-of-concept of a new method which has the potential to produce a more sensitive signal region in a shorter time than manual optimization, while keeping the analysis and interpretability as simple as possible.

This work builds on multi-dimensional histograms as implemented in the SparkDensityTree library [1], following [2–4]. SparkDensityTree has the following advantages compared to other density estimation methods.

Firstly, unlike most density estimation methods, including various regularization and Bayesian methods based on the likelihood, the minimum distance estimate (MDE) returned by SparkDensityTree as a multidimensional histogram is guaranteed to be within an $L_1$ distance or integrated absolute distance bound from the unknown density $f$ for any given sample size $n$, no matter what the underlying density $f$ happens to be, i.e., any density in $L_1$, the space of Lebesgue integrable functions, and is thus said to have *universal performance guarantees* [3].

Secondly, the method scales to arbitrarily large sample sizes in high dimensions due to a scalable implementation with sparse binary trees for representing the data. A sparse binary tree can represent only the existing data in its leaves by implicitly encoding the unrepresented leaves without data as zero akin to sparse vectors and matrices.

Thirdly, SparkDensityTree provides various further statistical insights from the MDE. In particular, calculating the coverage or highest density regions of the MDE histogram of the signal and background data allows for finding the region of phase space with the largest probability density in the signal and background. The highest density region covering the sample space for a given probability $1 - \alpha$, should have the smallest possible volume such that every point inside the region should have a probability density at least as large as every point outside the region. The method takes measured or simulated data for signal or background processes as input and returns the highest density region of its density estimate (MDE histogram). The signal region is given as a union of intervals, rectangles, cuboids and hyper-cuboids over the domain of the input variables.

The current proof-of-concept is largely inspired by a search for dark mesons in ATLAS data [5]. This search explores prompt decays of dark pions and dark rho mesons into standard model particles in LHC data. The data and simulation used in the following sections, as well as the selections applied, loosely follow the analysis. The full ATLAS analysis is however quite complex, and so, comparing directly with this work is not completely possible. The signal point chosen in this study has already been excluded by ATLAS [5], and so the data will be used as background.

## 2 Datasets and event selection

The dark sector signals searched by ATLAS [5] produce two final states: $tttb$ and $ttbb$. The one lepton final state is the most sensitive and it is characterized by events with one lepton and from 6 to 8 jets where at least 4 are identified as coming from the decay of a b-quark.

The study uses $2.3\,\text{fb}^{-1}$ of $\sqrt{s} = 13\,\text{TeV}$ proton–proton ($pp$) collision data collected by the CMS experiment [6] in 2015 to model the background to the dark meson signal. The analyzed data correspond to the SingleElectron [7] and SingleMuon [8] datasets released on the CERN Open Data portal [9]. Only events in the list of validated runs [10] are retained for the study. A total of about 110 million single electron and 70 million single muon $pp$ events are available for analysis.

The datasets are provided in the CMS miniAOD format, which contains high-level reconstructed objects that can be used for analysis [11]. This study is based on such reconstructed electrons, muons and jets. The data is accessed and processed using the CMS analysis code provided with the CMS open data [12]. Within this framework, jets are reconstructed using the anti-$k_t$ algorithm [13] with a fixed radius parameter $R = 0.4$ and are tagged as containing a bottom hadron (b-tagged) based on the Combined Secondary Vertex (CSV) tagging algorithm. In a complete data analysis, the fact that b-tagging efficiencies can differ significantly between data and simulation, has to be taken into account. This is however not critical for this study and so, no dedicated strategy has been implemented to address this.

The matrix element calculation for the dark meson production is performed at next-to-leading order (NLO) in QCD based on the model described in [14], using the corresponding UFO model files [15]. A signal sample is simulated using MadGraph5_aMC@NLO 3.5.1 [16] interfaced with Pythia 8.306 [17] for the modeling of parton showering, hadronization and underlying event with the A14 set of tuned parameters [18] and the NNPDF2.3lo [19] set of parton distribution functions. Fast simulation of the detector is done with Delphes 3.5.0 [20] using the standard CMS detector card.

Within Delphes, jets are determined with the FastJet 3.3.4 [21] software package and the anti-$k_t$ algorithm [13]. The default $b$-tagging of the CMS Delphes card is used to identify $b$-jets. The dark pion mass is set to $m_{\pi_D} = 500\,\text{GeV}$ and the dark rho mass to $m_{\rho_D} = 2\,\text{TeV}$. The signal cross-section is extracted from MadGraph5_aMC@NLO 2.9.9 [16] and amounts to 18.9 fb. As previously mentioned, this signal point has been excluded by the ATLAS collaboration [5]. A total of 50k signal events are simulated and the sample is normalized to the integrated luminosity of the data sample.

Events are further selected for the study based on kinematic and quality criteria imposed on the reconstructed leptons and jets. In the MC events, any electron or muon with transverse momentum $p_T > 28\,\text{GeV}$ is considered as a signal lepton. In data events, the signal lepton must additionally pass the *Tight* selection criteria [22,23]. Only events containing exactly one signal lepton are retained for the study.

All jets are required to have a transverse momentum $p_T > 20\,\text{GeV}$ and to satisfy $|\eta| < 2.5$. In addition, any jet is required to have an angular distance $\Delta R > 0.4$ from the signal lepton in the event, in order to resolve any reconstruction ambiguities between the lepton and jets. If these requirements are not met, the jet is discarded. Events are eventually required to have at least four jets, out of which at least two must be $b$-tagged.

Events passing all requirements listed here are selected for analysis. A total of 120k and 6.47 events pass this baseline selection in data and signal respectively.

## 3 Discriminating variables

The method is demonstrated on four event-level quantities that are suitable as discriminating variables, which are furthermore the same as the ones in [5]. The first three are; $\Delta R(l, b_2)$, defined as the angle between the lepton and the second closest $b$-jet; $m_{bb_{\Delta R_{min}}}$, defined as the invariant mass of the two $b$-jets in the event that are closest to each other; and $H_{\mathrm{T}}$, defined as the scalar sum of the $p_{\mathrm{T}}$ of the jets in the event. The final variable is based on $R = 1.2$ jets reclustered from the $R = 0.4$ jets using the anti-$k_t$ algorithm with a fixed radius parameter of $R = 1.2$ [24]. The lepton is added to the $R = 0.4$ jet collection before the reclustering and the highest-$p_{\mathrm{T}}$ large-$R$ jet containing the lepton is referred to as $\mathbb{J}^{lep}$ while the highest-$p_{\mathrm{T}}$ fully hadronic large-$R$ jet is referred to as $\mathbb{J}^{had}$. The sum of the masses of these two jets is used as a discriminating variable and is denoted by $m_{\mathbb{J}^{had}} + m_{\mathbb{J}^{lep}}$. Distributions of the discriminating variables in data and signal are shown in Figure 1 for events passing the baseline selection described in the previous section.

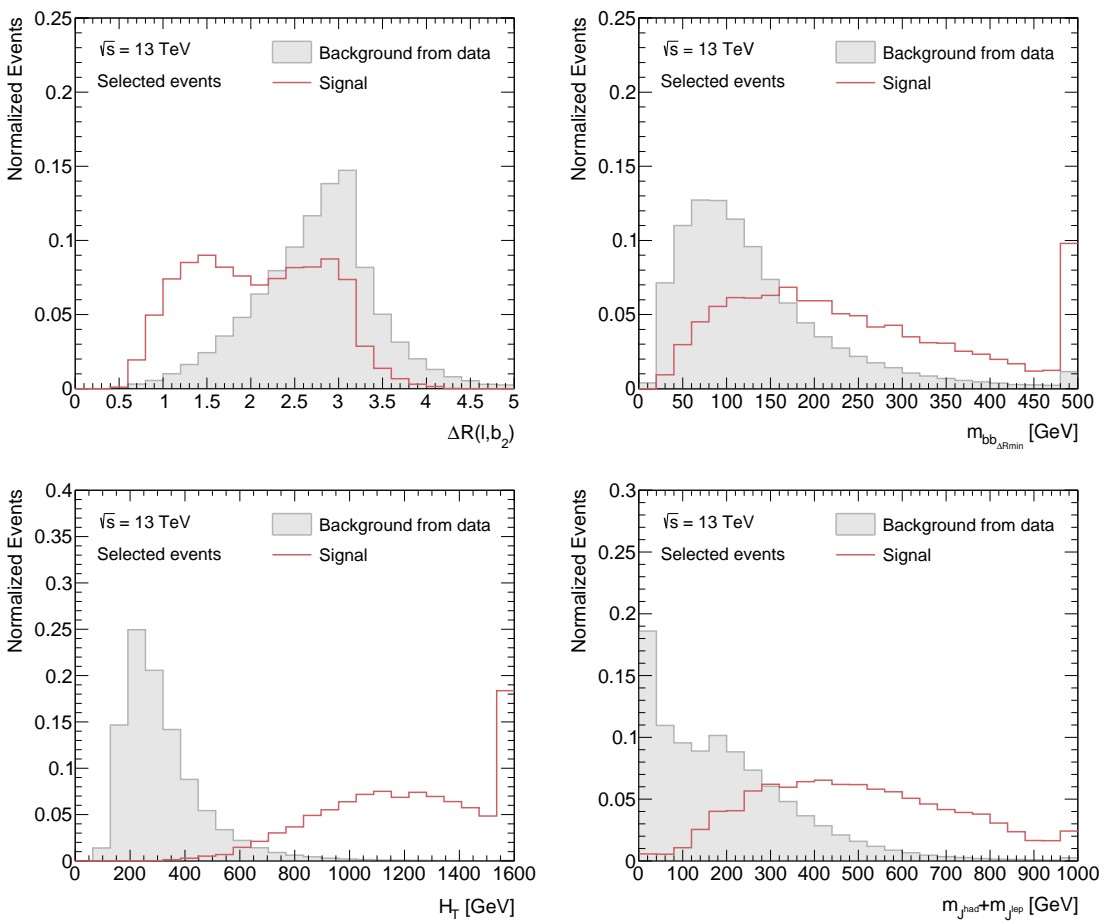

Figure 1: Distributions of the discriminating variables in data and signal for selected events, normalized to 1. (Top, Left): $\Delta R(l, b_2)$; (Top, Right): $m_{bb_{\Delta R_{min}}}$; (Bottom, Left): $H_{\mathrm{T}}$; and (Bottom, Right): $m_{\mathbb{J}^{had}} + m_{\mathbb{J}^{lep}}$.

# 4 Method

## 4.1 Mathematical description

The SparkDensityTree library is a library of statistical methods, with the base class being a multi-dimensional density estimator that for any sample generated from an unknown density returns an optimally smoothed histogram. The optimally smoothed histogram is taken to be the one that, per estimation, minimizes the $L_1$ distance to the true underlying distribution, using the MDE method. The statistical methods on these MDE histograms include arithmetic operations, conditional densities, coverage regions, and marginal densities. Recall that the marginal probability density of a subset of variables is obtained by integrating out all other variables in the joint probability density of all the variables.

The histogram object is represented as a binary tree in which each node represents a bisection of the phase space, and the leaves contain the event count in the finest resolution boxes thus obtained. The histogram construction begins with the definition of the root box, ideally the smallest hypercube containing all data points. From the root box, $\mathbf{x}_\rho$ - where $\rho$ represents the index of the widest coordinate, the support is iteratively bisected until a stopping criterion is reached, as visualized in Figure 2. The underlying tree structure [2] allows for assigning each box in the splitting a unique address. The combination of the leaf address and the counts is defined as the label of the box, $(\rho v, \#\mathbf{x}_{\rho v})$, where $v$ now represents the splitting direction. A two-dimensional example of this can be seen in Figure 2.

The MDE histogram is described in [3], and is taken as the optimal density estimate in this work. This quantity, $f_n(x)$, is defined as following [3]:

$$f_n(x) = \frac{1}{n} \sum_{\rho v \in \mathbb{L}(\mathbf{x}_\rho)} \frac{\#\mathbf{x}_{\rho v}}{\mathbf{vol}(\mathbf{x}_{\rho v})} \mathbb{I}_{\mathbf{x}_{\rho v}}(x), \tag{1}$$

where the volume, $\mathbf{vol}(\mathbf{x}_{\rho v})$, of a $d$-dimensional box is defined in detail in [2], $\mathbb{L}$ corresponds to the full set of leaves, $\mathbb{I}_{\mathbf{x}_{\rho v}}(x)$ is an identity matrix of the same dimension as the leaf address to retain dimensionality, and $n$ is the amount of data points in the root box. This quantity is found by an adaptive search in sequentially coarser histograms, starting at the one obtained by the splitting.

The splitting is an inherently sequential process, but a distributed solution was developed in [4, 25]. This requires an initial splitting of the root box down to the finest resolution that might be needed instantaneously – possibly to the point that each leaf only has a count of one – and then merged again. This is accomplished by only representing the leaves with at least one data point using sparse binary trees.

In the distributed method, therefore, an additional step is added between the splitting and the MDE, which consists of merging the cells to a stopping criterion on the counts in each box, effectively representing the initial histogram for finding the MDE.

For a more in-depth explanation of the steps, the reader is referred to [2–4, 25, 26]. The procedure is sketched below:

Stage 1: Find the root box containing all the data points.

Stage 2: Define a stopping criterion for the splitting, such as a maximum box size. The root box is split until this criterion is reached, giving the *finest resolution histogram*. In this work, the finest splitting is determined by the stopping criterion that no leaf-box has any side length longer than the parameter **finestResSideLength**.

Stage 3: Merge leaves such that the counts are maximized, while not going higher than some limit **minimumCountLimit** and keeping the leaf depth as small as possible.

Stage 4: Starting from the histogram obtained in stage 3, find the optimally smoothed histogram using MDE as described in [25, 26].

Additionally, two user-defined parameters concerning the distributed aspect of the method are available: **numTrainingPartitions** and **sampleSizeHint**. Respectively, they correspond to how many times the training data is partitioned, related to distribution of work among computing nodes, and an initial guess of points connected to the size of the node batches [27].

## 4.2 Implementation

The value of this method for data exploration in high-energy physics lies in the next step. When the MDE histogram is obtained, the highest density regions can be extracted by calculating the probability density function (pdf) coverage regions; and accordingly the highest and lowest density regions.

For simplicity, marginal densities are considered in this work, but the method can be extended to take the full density into account simultaneously.

The marginal densities for all unique pairs of the variables can be obtained from the 4-dimensional MDE histogram. In this paper, $\binom{4}{2} = 6$ unique pairs of variables are chosen and these six combinations are what the highest density regions are computed from. This is done separately for signal and background. The signal and background highest density regions can be defined independently of each other, and can, crucially, be flipped around to allow for finding the least dense region in the background density. From here, the user has to consider the best ways to use these marginal densities, and an example is given below.

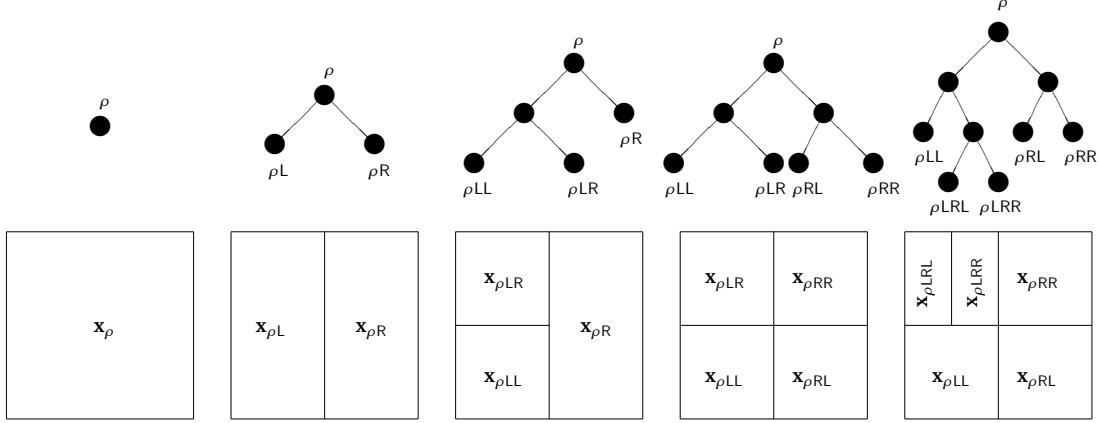

Figure 2: A sequence of splittings along the first widest coordinate, starting from the root box in two dimensions. Obtained from [2].

## 5 Results

Figure 3 shows a comparison between a 2D frequency or count histogram of the data over a uniform grid and that over the optimally smoothed nonuniform partition corresponding to the MDE histogram of this method. The main observation made from Figure 3 is that the distribution obtained from the method is similar enough to be considered adequate. Differences can be observed, such as the method not being able to fully capture the range of the distribution, but this is not seen as a major issue as the region of interest is contained. All distributions considered in this work have been compared in this way to ensure sensible density estimates are returned by the method.

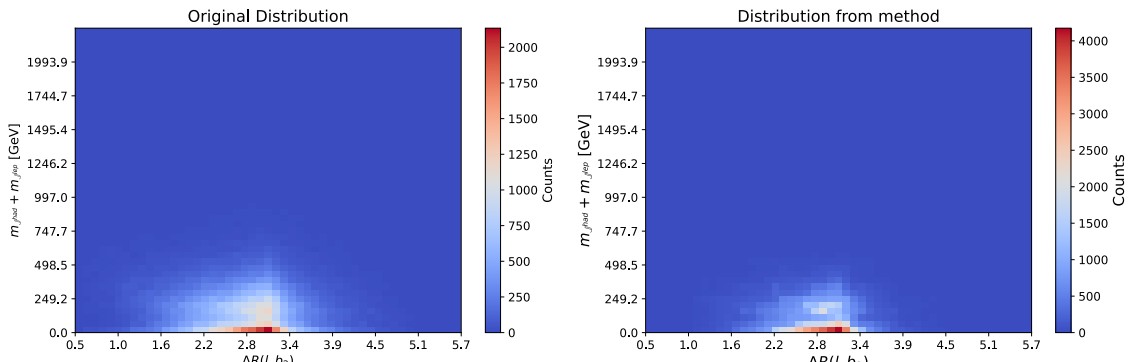

Figure 3: Comparison between a regular 2D histogram representation (Left) and the distribution obtained in this method (Right) of $m_{\mathbb{J}^{had}} + m_{\mathbb{J}^{lep}}$ vs. $\Delta R(l, b_2)$ in background from data.

The density estimate is presented at three different highest density regions for background in Figure 4, and for signal in Figure 5 for the $m_{\mathbb{J}^{had}} + m_{\mathbb{J}^{lep}}$ vs. $\Delta R(l, b_2)$ combination.

Comparisons between signal and background distributions can also be made at different levels. Figure 6 shows the 3D and 2D combinations, together with the highest 50% density regions for $m_{\mathbb{J}^{had}} + m_{\mathbb{J}^{lep}}$ and $H_{\mathrm{T}}$.

It is possible to see from Figures 4 and 5 that reducing the highest density region shrinks the area in which $f_n$ is non-zero - which is a sign that the method is working as expected. The main conclusion drawn from Figure 6 is that upon comparing the signal and background distributions, the amount of overlap decreases, implying an increase in discriminating power.

The density estimates for signal and background are combined to form $X_{\mathrm{sig}} \otimes \overline{Y}_{\mathrm{bkg}}$ density regions, where $X_{\mathrm{sig}}$ indicates the $X\%$ highest signal density region and $\overline{Y}_{\mathrm{bkg}}$ indicates the complement of the $Y\%$ highest background density region. These combinations are used to design kinematic regions corresponding to the most dense signal and the least dense background. The regions are achieved from the $X\%$ highest signal density region and the $Y\%$ highest background density region using a bounding box around the density region in each pair of variables. From the bounding box, the sensitive interval of each variable is taken as the projection of the box onto that axis. The intersection of the signal interval and the complement of the background interval forms the final interval of interest for each variable pair. Each variable is associated with exactly three intervals from its participation in three variable pairs. In this work, the final region is defined by the union of these intervals in each variable. Three combinations are presented: $50\% \otimes \overline{50}\%$, $90\% \otimes \overline{20}\%$ and $90\% \otimes \overline{10}\%$. As an example, the obtained intervals for the $90\% \otimes \overline{10}\%$ combination are:

$$\Delta R(l, b_2) : [0.6, 1.1] \, , \qquad\qquad H_{\mathrm{T}} : [625, 2172] \, \mathrm{GeV} \, ,$$
$$m_{bb_{\Delta R_{min}}} : [312, 634] \, \mathrm{GeV} \, , \qquad\qquad m_{\mathbb{J}^{had}} + m_{\mathbb{J}^{lep}} : [552, 996] \, \mathrm{GeV} \, .$$

When compared to the one-dimensional distributions in Figure 1 it is clear that these correspond to regions with discrimination power between signal and background. While direct comparison with the ATLAS analysis [5] cannot be done, since for example $m_{\mathbb{J}^{had}} + m_{\mathbb{J}^{lep}}$ is used as discriminant variable for the final fit, the intervals selected in this case are all fully contained in the signal region of the actual analysis.

The event selection corresponding to the intervals is applied to signal and data and the number of events passing the requirements are presented and compared in Table 1.

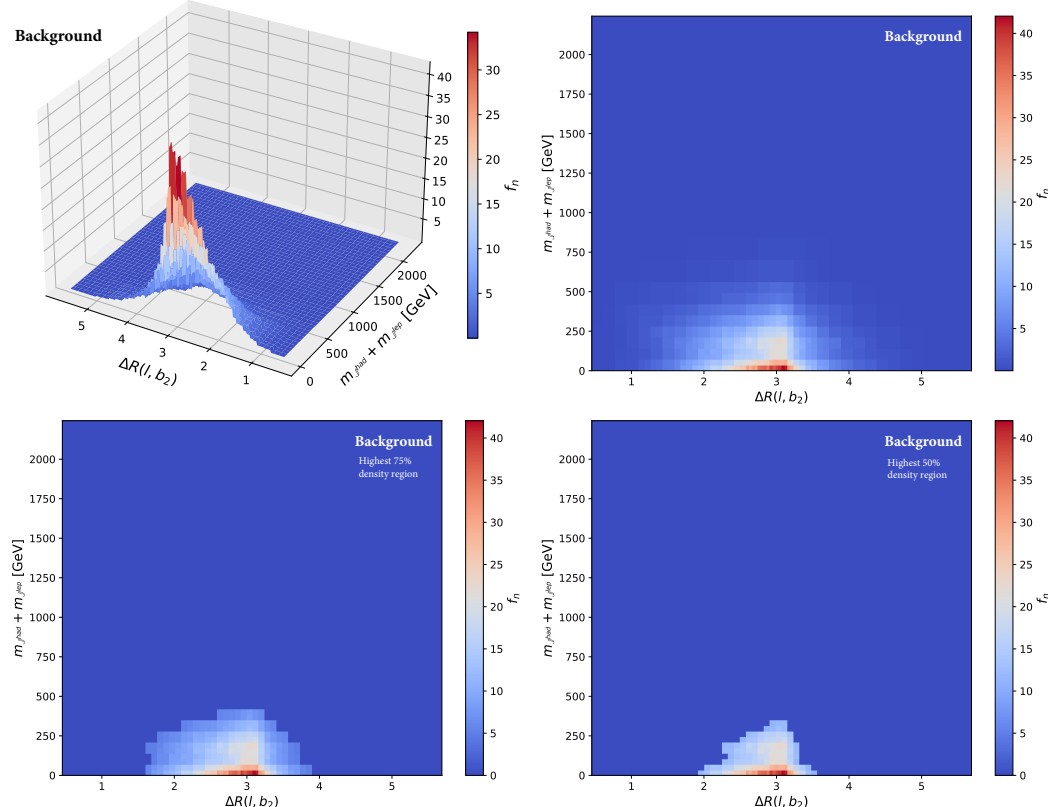

Figure 4: Full background density estimate of the $m_{\mathbb{J}^{had}} + m_{\mathbb{J}^{lep}}$ vs. $\Delta R(l, b_2)$ combination in 3D (Top Left) and 2D (Top Right) together with the highest 75% density region (Bottom Left) and the highest 50% density region (Bottom Right).

Table 1: Number of signal and background events passing the baseline analysis event selection and the selections derived from the density regions applied on top of the baseline. Relative numbers of events with respect to the baseline analysis selection are given within brackets. The signal numbers are normalized to the integrated luminosity of the dataset and the quoted uncertainties are statistical only. The last column shows the significance, defined as $s/\sqrt{s+b}$ where $s$ and $b$ are the number of signal and background events.

| Selection | Signal | | Background | | Significance |
|---|---|---|---|---|---|
| Baseline | 6.47 ± 0.07 | (100%) | 123950 ± 350 | (100%) | 0.018 |
| 50%⊗$\overline{50}$% | 0.565 ± 0.022 | (8.7%) | 364 ± 19 | (0.30%) | 0.030 |
| 90%⊗$\overline{20}$% | 0.295 ± 0.016 | (4.6%) | 16 ± 4 | (0.01%) | 0.073 |
| 90%⊗$\overline{10}$% | 0.072 ± 0.008 | (1.1%) | 0 ± 0 | (0.00%) | 0.27 |

The method results on less than one signal event on all tested scenarios and no background events pass the selections in the most aggressive selection. Dark meson signals are usually very small, and unlikely to be accessible in $2.3\,\text{fb}^{-1}$ of data. It is possible however to naively scale the 0.57 expected events in the 50%⊗$\overline{50}$% scenario to, e.g. the full Run 2 data set collected by ATLAS, containing $140\,\text{fb}^{-1}$, to more than 30 events, a reasonable signal for a new physics search.

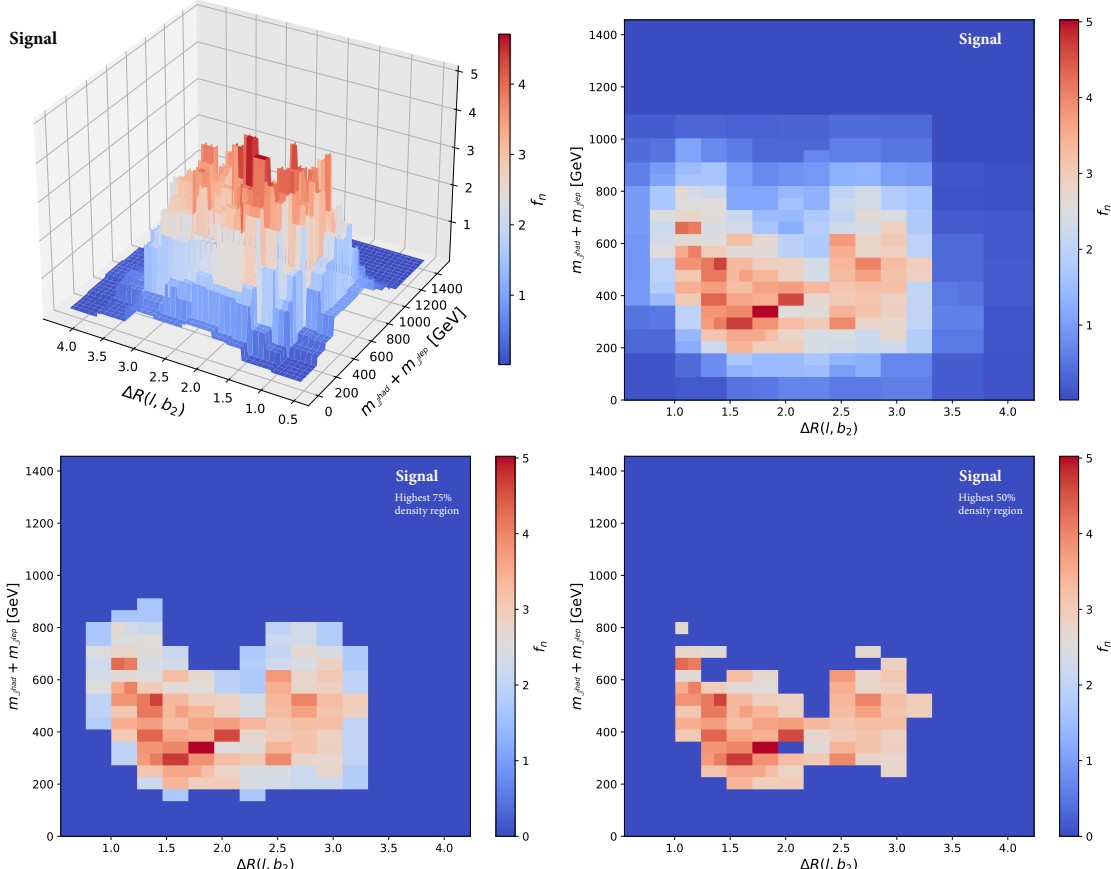

Figure 5: Full signal density estimate of the $m_{\mathbb{J}^{had}} + m_{\mathbb{J}^{lep}}$ vs. $\Delta R(l, b_2)$ combination in 3D (Top Left) and 2D (Top Right) together with the highest 75% density region (Bottom Left) and the highest 50% density region (Bottom Right).

The method could further be developed to identify the highest density region directly in the 4D histogram, and then project this onto the four axes. The SparkDensityTree library allows for defining arithmetic on the histograms, and it might be possible to combine the signal and background histograms and find the densest region in, e.g., number of signal events divided by number of background events, or the difference between the histograms.

Finally, scalability is a very powerful aspect of this approach. This study did not delve into it, but as mentioned in [4, 25, 26], the original method has been tested on several terabytes of simulations, and great decreases in computational time can be seen with the increase of cores. In this work, for example, the time required (averaged over 5 runs) to run over the CMS open data was 42.9 seconds, while the time required to run over the signal was 26.9 seconds.

The results presented in this work are documented in a Github repository [28]. All computations for the latter results have been performed on Virtual Machines (VMs) hosted by Google as a part of a dataproc cluster. The cluster contains three VM instances, all of which run four Intel Skylake vCPUs and has 15 GB of RAM; all in order to utilize the distributed aspect of the method.

This is something of interest for the field of high-energy physics, as it would be straightforward to run directly on the full collision datasets from the LHC.

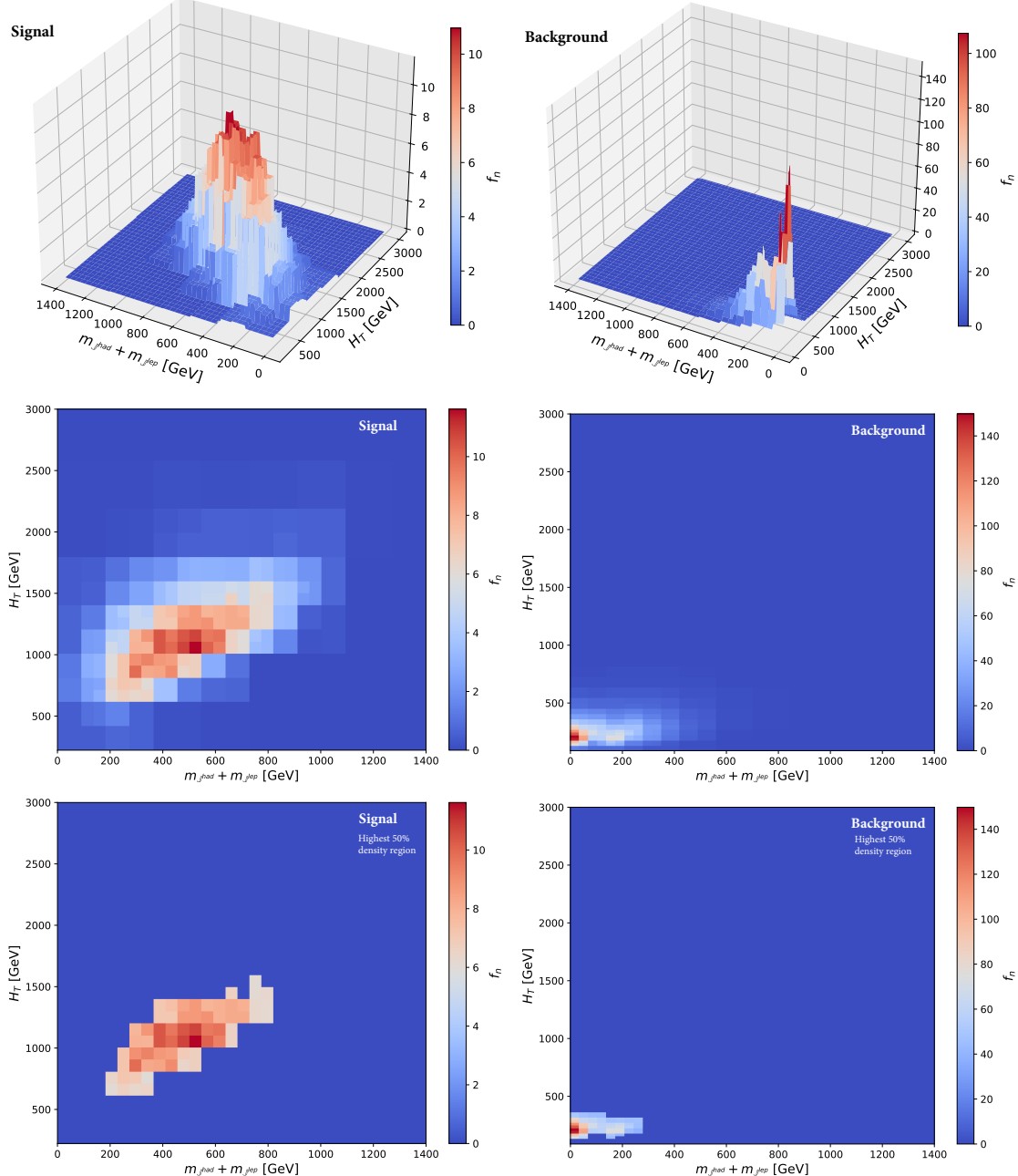

Figure 6: Full density estimate of the $m_{\mathbb{J}^{had}} + m_{\mathbb{J}^{lep}}$ vs. $H_{\mathrm{T}}$ combination in 3D (Top), 2D (Middle) and the highest 50% density region (Bottom) for signal (Left) and background (Right).

# 6 Conclusion and outlook

This paper introduces a scalable method, originally formulated in a purely mathematical context, applied for the first time in a high-energy setting. The approach relies on optimally smoothed multi-dimensional histograms with universal performance guarantees through scalable sparse binary tree arithmetic, incorporated in the SparkDensityTree library. It enables a rigorous definition of phase space regions enriched in signal, using multiple variables at a time. This method suggests promising avenues for the exploration of new physics phenomena at the LHC.

A large number of additional options is available from the SparkDensityTree library. This library contains several arithmetic operations and statistical methods (not covered here) that can be advantageous for studies on histograms, naturally interesting in a high-energy physics context.

## Acknowledgments

The authors want to thank the conveners and members of the Heavy Quarks and Top Subgroup in ATLAS for the interest in this project and the discussions.

**Author contributions**  The paper design and planning was done by OSG, and the method was developed by OSG and AG. Data preparation, Monte Carlo simulation, and analysis was done by GR and JJH. The manuscript was written by OSG, AG, GR, RS, and RGS. Final editorial work done by GR and RGS. Figures are from AG and GR. All authors participated in the discussion leading to this paper. The paper was approved by all authors.

**Funding information**  This research was partially supported by the project *AI4Research* at Uppsala University. This material is based upon work supported by the Google Cloud Research Credits program with the award GCP19980904. G. Ripellino is supported by the Carl Trygger foundation (CTS 20:1169). J. Heinrich is supported by the Department of Energy Office of Science Award DE-SC0017996. The Swedish Research Council supports A. Gallén and R. Gonzalez Suarez (VR 2023-03403). R. Sainudiin is partially supported by the Wallenberg AI, Autonomous Systems and Software Program funded by Knut and Alice Wallenberg Foundation. O. Sunneborn Gudnadottir is partially supported by the Centre for Interdisciplinary Mathematics (CIM) at Uppsala University.

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
