# Peer review of "Sparks in the Dark"

_SciPost Physics, doi:SciPost Phys. 18, 080 (2025)_

## Round 1 · Referee Report · Adrián Casais Vidal (Referee 1) · 2024-12-9

Strengths

1 - The text proposes a novel method to boost the intepretability for defining signal-enriched regions in the context of High Energy physics. I would like to stress that I find this idea very original and valuable.
2 - The paper provides a complete implementation of the method and code to support the explanation.
3 - This proof of concept opens a new family of methods that avoid the lack of interpretability that the state of the art ML methods have, with in principle little pay in performance.

Weaknesses

1 - The scalability in feature space might not be ideal in this vanilla implementation. The feature space presented in this paper is only 4-dimensional. Higher feature spaces might result in harder to handle computations. (The authors though address this point stating that they mitigated the effect).
2 - In exchange for interpretability, the application of this method requires the analyst to make more design choices with respect to the start of the art ML methods.
3- The samples used in the implementation example lack the granurality to show quantiatively the performance of the method.

Report

This work meets the elegibility criteria, agreeing with the authors as it checks two out of the four boxes:

- Provide a novel and synergetic link between different research areas.

The paper provides a solution to the long standing problem of defining a signal region in a novel way, providing interpretability with an alogirthmic approach.

- Open a new pathway in an existing or a new research direction, with clear potential for multi-pronged follow-up work.

It's definetely a new pathway, since this work builds on top of a mathematical framework that's only 5 years old. The authors establish that this is not the only interpretation of the general idea of providing interprebility, not even the only possible implementation. These last two points are intended to remark the wide alley of new work that could follow up.

This is out of the scope of the paper, but I would love to see how concrete implementations of the method compare to state of the art ML classification algorithms, both in terms of signal/background but also how the added interpretability is cruccial to the analysis.

At this stage I would suggest a small number of changes that in any case mean to showstop the publication but only would delaty it for a while, trying to improve the quality of the text.

As a final remark I want to stress that I'm impressed about the originality of the method and the new world of possibilities it can open.

Requested changes

Here are some comments that I have compiled after reading the text doing many passes so I apologise in advance if the comment order is not perfect.

1 - (Editorial) Use either the American or Enligsh rules across the text. Especial focus on the use of "s" and "z". I have spotted that in l.66 you change from British to American. Then on l. 71 again to British. Try to keep it uniform.
2- Figure1: the $\Delta$ symbol does not show up properly in the figure. Additionally you chose the label $\mathbb{J}$ (with a particularly stylised J) which is then not replicated in the figure label. This is not a big deal but maybe it makes it easier for the reader if you keep a simpler label that is easier to reproduce in the plot. Not a strong point, up to you.
3 - l138 "The underlying the tree structure" -> The underlying tree structure.
5 - l 138 "Giving each box" -> (something like) "assigning each box with ... "
6 -Section4 : The method section could be more clearly divided in two parts: the mathematical description and how it ca be used to define signal and background regions and the algorithmic implementation . First, I would separate this two parts. Some extra thoughts:
6.1 - I find a bit hard to read the mathematical summary. You show explicitely the calculation of Eq. (1) but not all the details about the mathematical framework have been presented. For example, what's $\mathbb{I}_{\mathbf{x}_{\rho v}}$? Why do you need to define the box $s$ with a new symbol, that then you never actually use? I find it useful that you expose the underlying equation to obtain the density estimator, but maybe I would rethink what details are necessary or how can you re-summarize it so that you don't need to define a lot of new symbols.
6.2 - Could you do two subsections? A first one stating the underlying mathematical idea taken from [3] , and more importantly, how did you cleverly use it to define the signal and background regions. Then there's a second one dwelving into the implementation details, which I think could be decoupled from this first bit. I hope this makes sense.
7 - (Editorial) l 173: Have you defined pdf before as in "probability density function (pdf)"?
8 - Figure 3: Could you zoom into the relevant region? There's a lot of 0 counts in the figure that difficult the comparison between the two histograms. The comment applies also to Figure 4.
9 - l 186 I would leave computational technicalities at the end of the section.
10 - Figure 2: do you have the right permissions to reproduce this figure?
11 - Table 1: I know this is just a summary of the outcome of the method but could you add statistic uncertainties to guide the eye as to how significantly does the S/B change with different cuts.
12 - (Editorial): l237: "time require" -> "time required". Don't add a new paragraph in l239 maybe?
13 - (Editorial) I might have missed other typos so I would do another thorough pass to spot new ones along the lines I've already mentioned.

Recommendation

Ask for minor revision

---

## Round 1 · Referee Report · Sergei Chekanov (Referee 2) · 2024-12-9

Strengths

2

Weaknesses

2 could be difficult to reproduce without a very simple tutorial

Report

The article "Sparks in the Dark" represents an interesting approach in searches for new physics using multi-dimensional approach. I'm especially pleased that the LHC open data were used for demonstration of the method. I believe SciPost is a perfect place for submission of such reports.

I did not find any significant issues with this submission.

Requested changes

None

Recommendation

Publish (easily meets expectations and criteria for this Journal; among top 50%)

---

## Round 1 · Referee Report · Anonymous (Referee 3) · 2024-12-17

Report

General assessment:
This paper presents a study on a new method to define signal regions in searches for new phenomena at hadron colliders. It uses a recent ATLAS dark meson search as inspiration, and the mathematical methodology implemented in the SparkDensityTree library. The final result shows a great potential in background mitigation with this method, which is of particular importance for signals which are rare/ave low cross-sections. This result is well motivated and holds importance for the particle physics community, and is of quality that meets the criteria of SciPost. The paper is well-written and generally clear. Below, I provide a few suggestions and questions for the authors' consideration. Thank you!

Questions:
- Why the choice of this CMS OpenData dataset specifically? Is it a matter of better suitability than the most recent datasets, or timeline between the release of OpenData and the completion of the work?
- How would systematic uncertainties factor in for this method, if used in a realistic analysis scenario? It could be beneficial to have a paragraph on the considerations that would be needed in this case. How systematic vs statistically dominated you expect your final analysis to be, if you perform either in a full Run-2 scenario or even a Run-2+Run-3 dataset scenario?

Comments on clarity of methodology and results:
- Some of the variables presented in section 4 are a bit obscure for someone not familiar with the work referenced. Could the variables $\rho$, $v$, and the $\mathcal{I}$ be shortly defined in the text?
- L138: "The underlying the tree structure" --> I don't understand this sentence; Do you mean "the underlying tree structure" (therefore a spurious "the") or the underlying something else in the tree structure (then the something else is missing)?
- Results section: There are a lot of plots which are just shown and referenced in the text, but without any explanation of what we see and why, and how we should read it. The reader would benefit from more guidance and description of those. For example:
-- Fig. 3: perhaps around L193, describe what are the main features and differences between the left and right plot. Are most of the events more contained in a smaller area on the right? What are the empty bins around the bulk of the events, what does it mean?
-- Describe what are the main features and conclusions you get from Figs. 4, 5, 6. Discuss more details of your result.
- Table I: It may be beneficial to add a third column with a significance estimate, such as s/$\sqrt{b}$, so the reader does not have to make that calculation themselves.

Comments on wording/typography/cosmetics of figures:
- Figure 1: top left and top right: there is a typographical mistake where a square box is shown instead of a $\Delta$ symbol for the two variables in the x axes of those plots.
- L143: where the volume --> where the volume (vol($x_{\rho v}$))
- Fig 4, 5, 6: Add some legend in the figure itself making it more clear it is the background (fig. 4) or signal (fig. 5), perhaps also write on the plot itself the highest 50%, highest 75% information; Add signal in the left side and background on the right side of the plots in Fig. 5; These makes looking back at your paper easier, without having to skim the details of the caption every time, and write on top of it ourselves as the reader.

References:
[5] Use "ATLAS Collaboration" instead of G. Aad et al, to be consistent with the formatting of other collaboration papers cited
[6] Same as above, use "CMS Collaboration"

Recommendation

Ask for minor revision

---

## Round 2 · Referee Report · Anonymous (Referee 1) · 2025-1-21

Report

I'm happy with the changes and the answers from the authors.

Recommendation

Publish (easily meets expectations and criteria for this Journal; among top 50%)

---

## Round 2 · Referee Report · Adrián Casais Vidal (Referee 2) · 2025-1-29

Report

Having read the newly updated version of the text and the compelling answers from the paper proponets, I'm now satisfied with the shape of the work here evaluated. I refer you to my original report for the breakdown of my criteria to eavaluate this research work, modulo the request changes.

As a unique and final remark I would have liked the minor editorial revision on the figures that I pointed out earlier. In any case, these are minor concerns and I focus on the main point, which is that I think this work is now ready for publication.

Recommendation

Publish (easily meets expectations and criteria for this Journal; among top 50%)

---

## Round 2 · Referee Report · Anonymous (Referee 3) · 2025-1-29

Report

Dear authors,
Thank you very much for your answers to the questions, and for the changes in the paper text and figures.
I am happy with the replies and changes.

Recommendation

Publish (easily meets expectations and criteria for this Journal; among top 50%)

---

## Round 2 · Author Response

We want to thank the editors for the comments and suggestions, we have implemented all of them and believe they improve the quality and readability of the paper.

---

## Round 2 · List of Changes

Editor 1: Questions: - Why the choice of this CMS OpenData dataset specifically? Is it a matter of better suitability than the most recent datasets, or timeline between the release of OpenData and the completion of the work?

Answer: CMS open data is used because 4 of the 6 authors are ATLAS affiliated, so using ATLAS open data would not be appropriate. Furthermore, at the time of writing, the CMS open data was more complete.

  • How would systematic uncertainties factor in for this method, if used in a realistic analysis scenario? It could be beneficial to have a paragraph on the considerations that would be needed in this case. How systematic vs statistically dominated you expect your final analysis to be, if you perform either in a full Run-2 scenario or even a Run-2+Run-3 dataset scenario?

Answer: This paper proposes a new method to perform the optimisation of the event selection for a typical search for new physics at a collider experiment. The goal is to find a phase-space region enriched in signal events. This enriched region allows for a statistical analysis that is sensitive to the signal. The statistical analysis itself, where the systematic uncertainties enter, is beyond the scope of our study. However, from the main result (https://arxiv.org/pdf/2405.20061, already public) we can see that the theoretical sources of uncertainty on the background modeling dominate the total uncertainty. There are some points in the signal grid that are still statistically limited, and those would benefit from Run 2 + 3.

Comments on clarity of methodology and results: - Some of the variables presented in section 4 are a bit obscure for someone not familiar with the work referenced. Could the variables “ρ”,”v” and “I” be shortly defined in the text?

Answer: We have added the following:

  • Description of ρ in L139: From the root box, xρ - where ρ represents the index of the widest coordinate, the support is iteratively bisected until a stopping criterion is reached, as visualized in Fig. 2.
    Description of v in L143/144: [...] , where v now represents the splitting direction. A two dimensional example of this can be seen in Fig.2. Description of I in L148: [...], I is an identity matrix of the same dimension as the leaf address to retain dimensionality, [...]

  • L138: "The underlying the tree structure" --> I don't understand this sentence; Do you mean "the underlying tree structure" (therefore a spurious "the") or the underlying something else in the tree structure (then the something else is missing)? Should be “The underlying tree structure”. It has been fixed in the text.

  • Results section: There are a lot of plots which are just shown and referenced in the text, but without any explanation of what we see and why, and how we should read it. The reader would benefit from more guidance and description of those. For example:

-- Fig. 3: perhaps around L193, describe what are the main features and differences between the left and right plot. Are most of the events more contained in a smaller area on the right? What are the empty bins around the bulk of the events, what does it mean?

The following has been added to the text: The main observation made from Figure 3 is that the distribution obtained from the method is similar enough to be considered adequate. Differences can be observed, such as the method not being able to fully capture the range of the distribution, but this is not seen as a major issue as the region of interest is contained

-- Describe what are the main features and conclusions you get from Figs. 4, 5, 6. Discuss more details of your result.

The following has been added to the text: It is possible to see from Figures 4 and 5 that reducing the highest density region shrinks the area in which $f_n$ is non-zero - which is a sign that the method is working as expected. The main conclusion drawn from Figure 6 is that upon comparing the signal and background distributions, the amount of overlap decreases, implying an increase in discriminating power.

  • Table I: It may be beneficial to add a third column with a significance estimate, such as s/√b, so the reader does not have to make that calculation themselves.

Answer: We have added a third column with the significance defined as s/sqrt(s+b)

Comments on wording/typography/cosmetics of figures:

  • Figure 1: top left and top right: there is a typographical mistake where a square box is shown instead of a Δ symbol for the two variables in the x axes of those plots. Answer: Well spotted, we have fixed that
  • L143: where the volume --> where the volume (vol(xρv)) Answer: Fixed

  • Fig 4, 5, 6: Add some legend in the figure itself making it more clear it is the background (fig. 4) or signal (fig. 5), perhaps also write on the plot itself the highest 50%, highest 75% information; Add signal in the left side and background on the right side of the plots in Fig. 5; These makes looking back at your paper easier, without having to skim the details of the caption every time, and write on top of it ourselves as the reader. Answer: We have added legends to the figures.

References: [5] Use "ATLAS Collaboration" instead of G. Aad et al, to be consistent with the formatting of other collaboration papers cited [6] Same as above, use "CMS Collaboration”

Answer: Both citations have the right collaborations in their entries to the bib file but since they are journal publications they follow the same format as others -and the collaboration is not shown. We are not familiar with SciPost preferences here and leave it then to the editor to change the style.

Editor 2:

1 - (Editorial) Use either the American or Enligsh rules across the text. Especial focus on the use of "s" and "z". I have spotted that in l.66 you change from British to American. Then on l. 71 again to British. Try to keep it uniform.

Answer: We have unified the text to American english.

2- Figure1: the Δ symbol does not show up properly in the figure. Additionally you chose the label J (with a particularly stylised J) which is then not replicated in the figure label. This is not a big deal but maybe it makes it easier for the reader if you keep a simpler label that is easier to reproduce in the plot. Not a strong point, up to you.

Answer: We have fixed the delta but it is not possible for us to also have the same J, and the author that defined it is no longer in the field.

3 - l138 "The underlying the tree structure" -> The underlying tree structure.

Answer: Fixed

4 - l 138 "Giving each box" -> (something like) "assigning each box with ... "

Answer: Fixed

5 -Section4 : The method section could be more clearly divided in two parts: the mathematical description and how it can be used to define signal and background regions and the algorithmic implementation . First, I would separate these two parts. Some extra thoughts:

Answer: We have separated this section in two as suggested.

5.1 - I find it a bit hard to read the mathematical summary. You show explicitly the calculation of Eq. (1) but not all the details about the mathematical framework have been presented. For example, what's Ixρv? Why do you need to define the box s with a new symbol that you never actually use? I find it useful that you expose the underlying equation to obtain the density estimator, but maybe I would rethink what details are necessary or how you can re-summarize it so that you don't need to define a lot of new symbols.

Answer: Correct that s was redundant to include, so this has been removed to avoid confusion. Description of Ixρv added in L148: [...], Ixρv is an identity matrix of the same dimension as the leaf address to retain dimensionality, [...]

5.2 - Could you do two subsections? A first one stating the underlying mathematical idea taken from [3] , and more importantly, how did you cleverly use it to define the signal and background regions. Then there's a second one delving into the implementation details, which I think could be decoupled from this first bit. I hope this makes sense.

Answer: This has been implemented in this version of the paper.

6 - (Editorial) l 173: Have you defined pdf before as in "probability density function (pdf)"?

Answer: This had not been done, but is now fixed. Well spotted!

7 - Figure 3: Could you zoom into the relevant region? There's a lot of 0 counts in the figure that make the comparison between the two histograms difficult. The comment applies also to Figure 4.

Answer: We cannot do that anymore unfortunately, but following suggestions from another referee, we have added legends that we hope help to improve clarity.

8 - l 186 I would leave computational technicalities at the end of the section.

Answer: Paragraph moved to L248

9 - Figure 2: do you have the right permissions to reproduce this figure?

Answer: Yes, one of the authors is an author of that paper as well.

10 - Table 1: I know this is just a summary of the outcome of the method but could you add statistical uncertainties to guide the eye as to how significantly does the S/B change with different cuts.

Answer: We have now added the statistical uncertainties

11 - (Editorial): l237: "time require" -> "time required". Don't add a new paragraph in l239 maybe? Answer: Fixed

12 - (Editorial) I might have missed other typos so I would do another thorough pass to spot new ones along the lines I've already mentioned.

Answer: We have gone through it to the best of our ability

Editor 3: No comments

---

## Editorial Decision

published